# Geometry symmetry-free and higher-order optical bound states in the continuum

Qingjia Zhou[1,2], Yangyang Fu [3✉], Lujun Huang [4], Qiannan Wu[5], Andrey Miroshnichenko [4], Lei Gao [1,2✉] & Yadong Xu [1,2,6✉]

Geometrical symmetry plays a significant role in implementing robust, symmetry-protected, bound states in the continuum (BICs). However, this benefit is only theoretical in many cases since fabricated samples' unavoidable imperfections may easily break the stringent geometrical requirements. Here we propose an approach by introducing the concept of geometrical-symmetry-free but symmetry-protected BICs, realized using the static-like environment induced by a zero-index metamaterial (ZIM). We find that robust BICs exist and are protected from the disordered distribution of multiple objects inside the ZIM host by its physical symmetries rather than geometrical ones. The geometric-symmetry-free BICs are robust, regardless of the objects' external shapes and material parameters in the ZIM host. We further show theoretically and numerically that the existence of those higher-order BICs depends only on the number of objects. By practically designing a structural ZIM waveguide, the existence of BICs is numerically confirmed, as well as their independence on the presence of geometrical symmetry. Our findings provide a way of realizing higher-order BICs and link their properties to the disorder of photonic systems.

[1] School of Physical Science and Technology & Collaborative Innovation Center of Suzhou Nano Science and Technology, Soochow University, Suzhou, China. [2] Jiangsu Key Laboratory of Thin Films, Soochow University, Suzhou, China. [3] College of Science, Nanjing University of Aeronautics and Astronautics & Key Laboratory of Aerospace Information Materials and Physics (NUAA), MIIT, Nanjing, China. [4] School of Engineering and Information Technology, University of New South Wales, Canberra, ACT, Australia. [5] School of Science, North University of China, Taiyuan, Shanxi, China. [6] State Key Laboratory of Functional Material for Informatics, Shanghai Institute of Microsystem and Information Technology, Chinese Academy of Sciences, Shanghai, China. ✉email: yyfu@nuaa.edu.cn; leigao@suda.edu.cn; ydxu@suda.edu.cn

Recently, bound states in the continuum (BICs) have attracted a growing interest in the optics community[1–7], owing to their fundamental properties and their practical applications, such as strong resonances[8–10] and high-quality optical lasing[11–13]. BICs are unique waves that lie in the continuum but remain entirely confined without any radiation. BICs also features a resonance with an infinite quality ($Q$) factor in a spectrum, i.e., they are dark modes. Photonic BICs originating from various mechanisms have been found in a number of optical systems, such as photonic crystal slabs[3,14], dielectric gratings[15,16], spheres/rod arrays[17,18], waveguides[19,20], and others[21]. In particular, geometrical symmetries play a significant role in the study of photonic BICs, resulting in the so-called symmetry-protected BICs that may be found in several systems[14,19,22]. A simple example is provided by a two-dimensional (2D) waveguide structure loaded with two identical objects placed symmetrically in space, see Fig. 1a. Owing to reflection symmetry, the coupling of the discrete bound state of one symmetry class (i.e., odd symmetry) to the other symmetry class's continuous spectrum (i.e., even symmetry) is forbidden, thus leading to the existence of symmetry-protected BICs. Physically, electromagnetic (EM) guide modes supported by the waveguide may be described by the scalar wave function $\psi_m(x, y)$, which obeys the Helmholtz equation: $\hat{H}\psi_m(x, y) = (\omega_m^2/c^2)\psi_m(x, y)$, where $\hat{H} = \nabla^2 + [n^2(x, y) + 1]\omega_m^2/c^2$ is the Hamiltonian, $n(x, y)$ is the spatially dependent refractive index profile, $\omega$ is the frequency and $c$ is the light speed. The symmetry-protected BICs are ensured by the commutation relation $[P_y, \hat{H}] = 0$, where the parity operator $P_y$ is defined by the transformation $y \rightarrow -y$, and the equality $n(x, y) = n(x, -y)$ is guaranteed by the reflection symmetry.

The advantage of geometrical-symmetry-protected BICs is their robustness. However, this advantage often comes at a cost: the geometry of the system should be precisely controlled to ensure the symmetry of continuous waves and bound states in the system. A slight deviation from the required geometric structure may break the symmetry, leading to $[P_y, \hat{H}] \neq 0$ and, in turn, to the loss of robustness. In practice, deviations from the exact symmetry are unavoidable, due to imperfections in the fabrication technology. This is true in particular at higher operating frequencies, such as THz or in the visible spectrum. As a result,

the advantage of robustness due to geometrical symmetry is easily lost and it is only of theoretical significance. It is thus very relevant to investigate any mechanism leading to symmetry-protected BICs that are insensitive to the system's geometry. Although topological photonic crystals have been previously proposed to realize topological Fano resonances (i.e., quasi-BICs) robust to geometrical imperfections[23], the topological photonic crystals themselves are governed by the symmetry of geometric lattice, leading to similar control problems on the system's geometry. Therefore, it is significant to achieve robust BICs beyond the limit of geometrical symmetry and how to achieve robust BICs in a disordered system is still an open question.

Due to their static-like field distribution, zero-index metamaterials (ZIMs) make it possible to implement numerous optical phenomena[24,25], such as squeezing wave energy[26,27], tailoring wavefront[28–30], total transmission and reflection[31,32] and a few others[33–35]. In particular, the concept of photonic doping[36] has been proposed to tailor the effective material parameters of a 2D ZIM host doped with macroscopic dielectric objects. In fact, the effective permeability $\mu_{\text{eff}}$ of the doped ZIM may be significantly modified, while maintaining its effective permittivity unchanged (i.e., $\varepsilon_{\text{eff}} \cong 0$) for traverse magnetic (TM) waves. Regardless of the location, size, and number of the doping objects, the composite structure is equivalent to a uniform system with a constant index profile (i.e., $C = \sqrt{\varepsilon_{\text{eff}}\mu_{\text{eff}}}$). Therefore, although a doped ZIM may have no geometric symmetry, the condition $[P_y, \hat{H}] = 0$ may be valid because of $\hat{H} = \nabla^2 + (C + 1)\omega_m^2/c^2$. In other words, the reflection symmetry is preserved by the physical ZIM homogenization at the macroscopic scale. Given $N$ identical objects immersed in a ZIM, the reflection symmetry holds even if they are placed at arbitrary random locations. This fact provides the basis to design a system with BIC based on physical symmetry rather than geometric one.

In this work, we introduce the concept of geometric-symmetry-free but physical-symmetry-protected BICs, i.e., a ZIM host embedded with $N \geq 2$ objects, which support radiative monopole modes with non-zero magnetic flux. In particular, we demonstrate the existence of robust geometric-symmetry-free BICs, which could be realized regardless of the specific positions, external shapes, and material parameters of the objects in the ZIM host, owing to the nontrivial zero value of the total effective magnetic flux in the $N$ objects. In addition, we find that using $N$ doping objects enables higher-order BICs, and derive an analytical formula for the $N$-dependent $Q$ factor. Specifically, the $N$ objects can produce ($N$-1)-fold degenerate BICs. These results are very different from those reported previously for ZIM-based optical systems[37–39], where the BICs (embedded eigenstates) are induced by the non-radiative higher-order cavities modes (e.g., dipole mode) with zero magnetic flux in each object. Our results break the conventional wisdom of a ZIM-based BIC excluded from monopole modes[40], and pave a way to study the higher-order BICs and the associated physics.

## Results

**Models and theory.** We start our discussion from a typical 2D waveguide system (Fig. 1a). The upper and lower boundaries of the waveguide are perfect electrical conductors (PECs). For TM polarization (the magnetic field is along the $z$-direction), the waveguide supports two continuous waves (guide modes), with even and odd symmetry with respect to the axis of symmetry of the waveguide (the dash dot line). Those odd modes have a cutoff frequency. If we introduce two identical objects with reflection symmetry, we obtain two bound modes localized near the objects, showing even and odd symmetry, respectively. The odd bound mode is embedded in even guided mode continuum, which

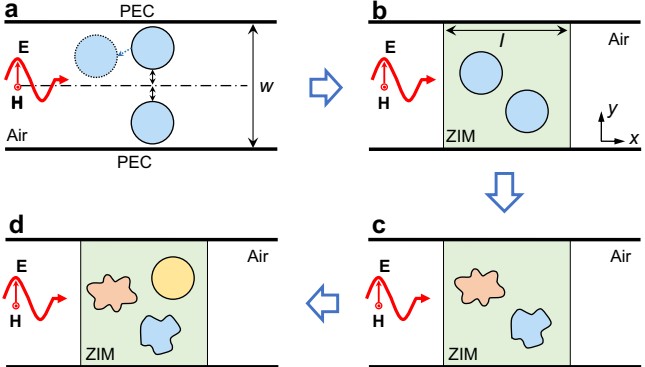

**Fig. 1 Geometric-symmetry-free and higher-order BICs induced by zero-index media (ZIM) in a waveguide. a** A two-dimensional (2D) waveguide structure with reflection symmetry for symmetry-protected BICs, where two identical objects (blue circle with solid boundary) are placed symmetrically in space. If one object is shifted (the blue circle with a dashed boundary), the BIC disappears. **b** After filling ZIM in the waveguide, the system supports a BIC which is independent of the location of objects. **c** The geometric-symmetry-free BIC is insensitive to the doping objects' shapes, materials, and distributions. **d** Higher-order BICs are implemented by embedding multiple objects in ZIM.

guarantees a robust BIC mode[1]. However, when one of the object is spatially shifted (the circle with dashed boundary in Fig. 1a), the system's reflection symmetry is lost, leading to the disappearance of the robust BIC mode. In the following, we will show that such BIC mode could be preserved by introducing the background medium of ZIM for the two objects (see Fig. 1b), even if the external shapes, materials, and number of these objects are arbitrarily changed (see Fig. 1c, d). Let us assume that $N$ objects of arbitrary shape are randomly placed in a ZIM host (Fig. 1d), and that they are non-magnetic dielectrics with permittivity $\varepsilon_d$. To uncover the underlying physics of BICs in this kind of systems, impedance-matched is assumed in the ZIM, i.e., $\varepsilon_1 = \mu_1 \to 0$. We consider a TM wave with an amplitude of 1 A m$^{-1}$ normally incident from the left. Following ref. [32], we have that the transmission coefficient of the ZIM waveguide system is given by

$$T = \frac{1}{1 - (i\omega/2wH_1)\sum_{i=1}^{N}\oint_{\partial C_i}\mathbf{A}_i \cdot d\mathbf{l}} \tag{1}$$

where $\mathbf{A}_i$ is the magnetic potential at the boundary $\partial C_i$ of the $i$-th objects, $w$ is the width of ZIM, $H_1$ is a local magnetic field in the ZIM, $\omega = 2\pi f$ is the angular frequency of the incident wave, and $f$ is the working frequency. The term $\oint_{\partial C_i}\mathbf{A}_i \cdot d\mathbf{l}$ represents the magnetic flux inside the $i$-th object. For the sake of simplicity, let us consider a cylindrical shape first (Fig. 1b). Then the magnetic flux inside the $i$-th cylinder can be expressed as

$$\varphi_i = \oint_{\partial C_i}\mathbf{A}_i \cdot d\mathbf{l} = \frac{2\pi H_1}{\omega}\frac{J_1(k_dR_i)R_i}{J_0(k_dR_i)\sqrt{\varepsilon_d}} \tag{2}$$

Here $k_d = \sqrt{\varepsilon_d}\omega/c$ is the wave vector in the objects and $R_i$ is the $i$-th cylinder's radius. From Eqs. (1) and (2), we see that the transmission coefficient depends on the properties of the object, but it is independent of the objects' location in the ZIM.

**Geometry symmetry-free BICs.** We first consider the simple case of two cylindrical objects in the ZIM (Fig. 1b). Figure 2a shows

the analytical results for the transmission coefficient as a function of $\varepsilon_d$ and $\alpha$, where $\alpha = (R_2 - R_1)/R_1$ is the asymmetry parameter. In the analysis, we fix the working frequency at 15.0 GHz by considering a narrow bandwidth of the ZIM and also keep the size of one object ($R_1 = 8$ mm), only change the permittivity of objects and the asymmetry parameter to observe the proposed BICs conveniently. As it is apparent from Fig. 2a, for a fixed $\alpha$, a transmission resonance emerges as $\varepsilon_d$ changes. In particular, for $\alpha \to 0$, such transmission peak becomes very sharp, and the corresponding quality factor diverges $Q \to \infty$. However, at $\alpha = 0$ the transmission peak turns into a dip. This is a typical signature of a BIC mode, i.e., a resonance with zero linewidth. Moreover, Fig. 2b shows the transmission spectrum for $\alpha = -0.01$, corresponding to a quasi-BIC mode. An electromagnetically induced transparency (EIT)-like behavior appears, with a peak occurring at $\varepsilon_d = 4.86$, and two valleys at $\varepsilon_d = 4.82$ and $\varepsilon_d = 4.92$, respectively. The typical feature of our system different from others is that our revealed quasi-BIC shares EIT-like lineshape, which is a special type of Fano resonance[41]. We validate our results by numerical simulations with objects at different locations, see red and yellow balls in Fig. 2b. The numerical results agree with the analytical ones, and the EIT-like behavior is preserved regardless of the two dielectric rods' locations. In simulations, the real part of the magnetic field is 1 A m$^{-1}$ for incident wave arriving at the left boundary of ZIM. The imaginary (real) part of magnetic field in the objects is dominant for the case of total reflection (total transmission). To clearly reveal the field enhancement and the phase relationship in the objects, the corresponding imaginary/real part of the magnetic field in the ZIM region of Fig. 1b is shown in Fig. 2c, d. Physically, these two valleys are stemming from monopole mode resonance occurring in either the object with $R_1$ (the left patterns in Fig. 2c, d) or the object with $R_2$ (the right patterns in Fig. 2c, d). Monopole mode resonances coincide for the two objects for the transmission peak, yet out of phase (the middle patterns in Fig. 2c, d). Note that such quasi-BIC mode leads to great field enhancement inside two objects: the EM field in the center is increased by >100 times compared to the incident field. Hence, the revealed (quasi)-BIC

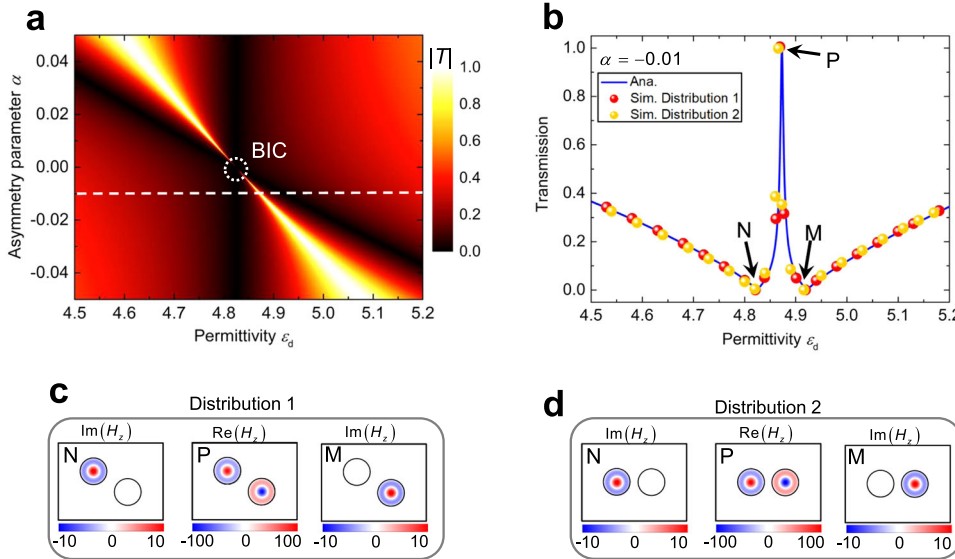

**Fig. 2 BIC illustrations. a** 2D map of the transmission coefficient as a function of the permittivity $\varepsilon_d$ and the asymmetry parameter $\alpha$. An ideal BIC occurs at $\alpha = 0$ and $\varepsilon_d = 4.82$ (the white dash circle). **b** Transmission spectrum for $\alpha = -0.01$. The solid blue curve denotes the analytical results. The red and yellow balls represent the simulated results for the two different distributions. The simulated magnetic field in the ZIM region for distribution 1 and distribution 2 is shown respectively in panels (**c**) and (**d**). N, M, and P correspond to the two transmission dips and the peak, respectively. In the numerical simulations, the incident magnetic field is 1 A m$^{-1}$, and the parameters of ZIM are $\varepsilon_1 = \mu_1 = 10^{-4}$. The other relevant parameters are: $R_1 = 8$ mm, $w = 44$ mm, $l = 60$ mm, and the working frequency is 15.0 GHz.

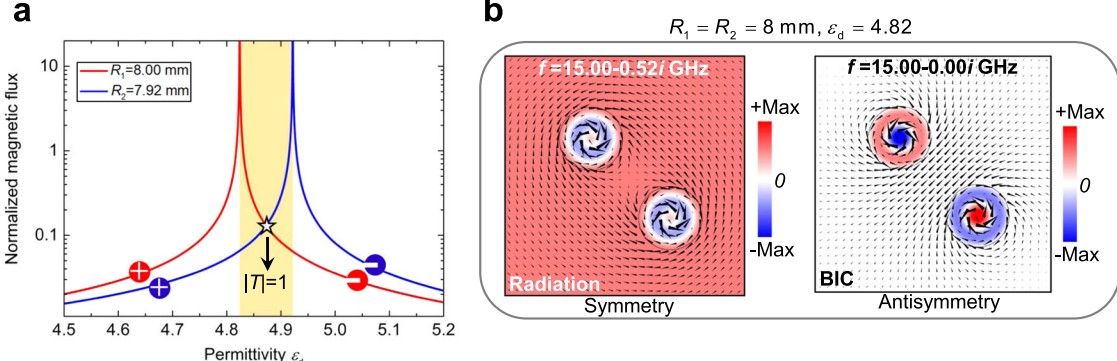

**Fig. 3 Normalized magnetic flux and eigenmode analysis. a** Normalized magnetic flux in each object as a function of $\varepsilon_d$, "+" and "−" denote the direction of magnetic flux. **b** Eigenmode analysis for a ZIM environment with two identical objects with $R_1 = R_2 = 8$ mm and $\varepsilon_d = 4.82$. Filling contours are out-of-plane magnetic field, and black cones represent the in-plane electric field.

mode provides an alternative way to enhance the optical non-linear response[42,43].

To reveal the underlying physical mechanism, responsible for BIC modes' appearance, we define a normalized magnetic flux for each objects, i.e., $\Phi_i = \varphi_i \omega/(2\pi H_1)$, where the normalized factor $\omega/2\pi H_1$ is the same for all the objects. Figure 3a shows $\Phi_i$ as a function of $\varepsilon_d$ for the two objects. For each object, the induced $\Phi_i$ changes with $\varepsilon_d$, featuring a strong resonance. The induced magnetic flux is pointing in opposite directions before and after this resonance, marked by the symbols "+" and "−" in the plot. Due to the different radii of two objects, their peaks do not overlap and are located at $\varepsilon_d = 4.82$ and $\varepsilon_d = 4.92$, respectively. These two values are consistent with the two transmission valleys in Fig. 2b. Between these two resonances, there is an intermediate region (the yellow area), in which the induced magnetic fluxes of these two objects have opposite directions. In particular, when the amplitudes of two magnetic fluxes are equal (i.e., at the crossing point), the total magnetic flux is zero, i.e., $\sum \Phi_i = 0$ while $\Phi_i \neq 0$. The permittivity corresponding to this crossing point is exactly the same as that for the transmission peak in Fig. 2b. For vanishing asymmetry parameter ($\alpha \to 0$), the two resonance peaks tend to overlap and the middle region shrinks. Yet, the total magnetic flux is always zero, thus ensuring the occurrence of the transmission peak. One may conjecture that for $\alpha = 0$ the two resonance peaks are going to coincide exactly at $\varepsilon_d = 4.82$, and that there are two-fold degenerate states in the two objects, i.e., one state with infinite total flux and the other one with zero total flux. In particular, the state with zero total flux is a dark mode, leading to ideal BIC. Thus, the BIC mode discussed above is physically related to the nontrivial zeros of the total magnetic flux, i.e., $\sum \Phi_i = 0$ while $\Phi_i \neq 0$, which distinguishes from the common situation of $\sum \Phi_i = 0$ with $\Phi_i = 0$[40]. In fact, $\sum \Phi_i = 0$ is general condition for BICs in the doped ZIM system and applies to two different situations, i.e., radiative monopole modes related to $\Phi_i \neq 0$, and higher-order cavities modes caused by zero magnetic flux in each object ($\Phi_i = 0$). In addition, it is noted that similar to the conventional symmetry-protected BIC, the proposed BIC can exist at any permittivity for the objects with a fixed size, as long as its shifted working frequency is still located in the frequency window of ZIM (see Supplementary Fig. 1).

In order to validate this picture, we analyze the eigenmodes of the doped ZIM numerically and analytically (see Supplementary Note 1). For $\alpha = 0$, there are two eigenmodes at the eigenfrequency 15 GHz, and Fig. 3b shows the corresponding field distributions, where the color contour is the out-of-plane magnetic field, and the black cones denote the directions of the in-plane electric field. Note that we cannot define exactly the

symmetry of the two eigenmodes due to the absence of any geometric symmetry. However, suppose we focus on the magnetic field distributions in the two objects. In that case, we can see that one of these two eigenmodes exhibits symmetry-like features, e.g. in-phase magnetic field profiles in the two objects, and vortex-like electric fields in the two objects having the same rotation directions. The other one is the anti-symmetric-like mode, as indicated by the out-phase magnetic field profiles in two objects, and by the electric fields' opposite rotation directions. Intuitively, these two-fold degenerate eigenmodes result from the interference of the fields inside the ZIM background radiated from monopole modes in the two objects, as they have identical resonant frequency and easily couple with each other. The anti-symmetry mode with zero total flux corresponds to a BIC mode (dark mode), which has a zero linewidth and is totally decoupled with external incidence, as reflected by the zero magnetic fields in the ZIM background (see Fig. 3b). As a result, the expected total transmission from the condition of zero total flux in Eq. (1) cannot appear (but it is accessible for a quasi-BIC mode with narrow linewidth as it is can couple with external incidence). Instead, the symmetry state (bright mode) with a non-zero background field (see Fig. 3b) is coupled with the external incidence, which leads to the zero transmission (see Fig. 2a) owning to its infinite total flux. Thanks to the quasi-static feature and constant field in the ZIM, these eigenmodes can usually exist independently of their specific distribution.

Moreover, as the BICs result from the nontrivial zeros of the total magnetic flux in the objects, it offers great flexibility for the material parameter and geometric shape of all the objects in ZIM (see Fig. 1c). To demonstrate these features, we numerically studied several cases of dielectric objects with square, triangle, arbitrary external shapes and different filling materials. In all cases, similar BICs are found, always coming from the nontrivial zero of the total flux in the two objects with the anti-symmetry-like mode state, which are displayed respectively in Supplementary Figs. 4–7. In particular, even in a more general model of two objects consisting of a rectangular object with $\varepsilon_d = 9$ and a cylinder with $\varepsilon_d = 4.82$, the proposed BIC is revealed analytically and numerically (see Supplementary Note 2 and Supplementary Fig. 8), which well confirms the existence of the proposed BIC independent of the material parameter and geometric shape of these objects in ZIM.

**Higher-order BICs**. Following the results obtained with two objects, we turn attention to the general case, i.e., a ZIM with an arbitrary number of objects ($N$ groups of objects). Interestingly, we find that the presence of $N$ objects leads to ($N$-1)-th order

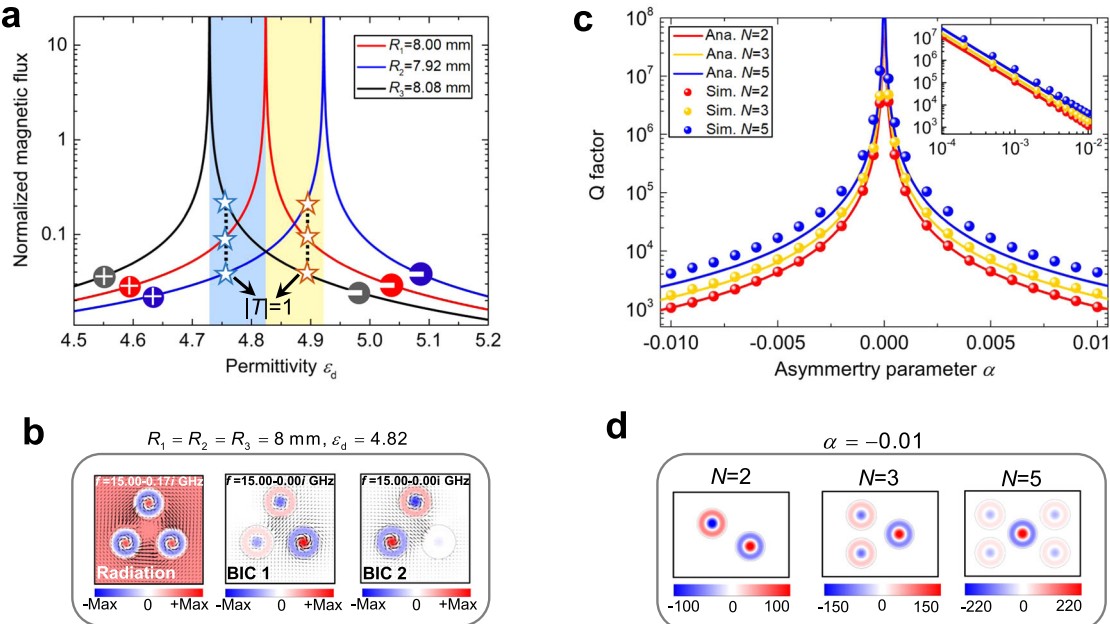

**Fig. 4 Higher-order optical BICs. a** Normalized magnetic flux as a function of $\varepsilon_d$ for a ZIM with three objects having $R_1 = 8\,$mm, $R_2 = 7.92\,$mm and $R_3 = 8.08\,$mm. **b** The eigenmodes for a ZIM host with three identical objects ($R_1 = R_2 = R_3 = 8\,$mm and $\varepsilon_d = 4.82$) at 15.0 GHz. **c** $Q$ factor as a function of the asymmetry parameter for different values of $N$ (the number of objects). The solid curves and circles denote the analytical and numerical results, respectively. The inset shows the same results in log scale. **d** Out-of-plane magnetic field at total transmission for $\alpha = -0.01$. Higher-order quasi-BICs are observed.

BIC. For simplicity, as shown in Fig. 1d, let us consider the case $N = 3$ to illustrate the phenomenon. The analytical and numerical results for transmission spectrum are shown in Supplementary Fig. 9, where the radius of the first object is fixed $R_1 = 8\,$mm, and the other ones have slightly different radii, i.e., $R_2 = 7.92\,$mm and $R_3 = 8.08\,$mm. As apparent from the plot, two EITs may be seen in the transmission spectrum, stemming from the zero total magnetic flux in the three objects, i.e., $\sum\Phi_i = 0$. Figure 4a shows $\Phi_i$ as a function of $\varepsilon_d$ for the three objects. There are three resonant peaks, which divide the whole range into four areas. There are two intermediate regions (colored regions in the plot), and there is a point where $\sum\Phi_i = 0$ in each region. This means there are two quasi-BIC modes in the $N = 3$ case. As $\alpha_2 \to 0$ and $\alpha_3 \to 0$, these two quasi-BIC modes coincide, leading to a high-order BIC mode. Further, the ZIM background's eigenmodes with three identical objects (corresponding to $\alpha_i = 0$) have been analyzed, and their corresponding field distributions are shown in Fig. 4b: three eigenmodes appear at the eigenfrequency 15 GHz. Unlike the $N = 2$ case, the symmetry of eigenmodes in this situation cannot be exactly defined, because the magnetic field distributions in three objects are irregular. Instead, these three modes can be distinguished by the intensity of the magnetic field in ZIM, i.e., one bright mode with a non-zero magnetic field in the ZIM, and two dark modes with zero magnetic field in the ZIM. In particular, both dark modes fully decouple from the external excitation and enable two-fold degenerate BIC. Strictly speaking, the symmetry of BICs in a general configuration cannot be defined, except for the case of $N = 2$. Accordingly, for $N$ objects in ZIM, there are $N-1$ groups of EITs and $N-1$ groups of quasi-BIC modes (linked to the condition $\sum\Phi_i = 0$). Consequently, $N$ identical objects in the ZIM induce $(N-1)$-fold degenerate BICs.

The presence of higher-order BIC is further confirmed by looking at the $Q$ factor for $N$ objects' case. For simplicity, we assume that the $N$ objects consist of $(N-1)$ identical objects with fixed radius $R_a = 8\,$mm and one object with a variable radius $R_b$.

After some calculations (see Supplementary Note 3), the $Q$ factor of the quasi-BIC is derived as,

$$Q = \frac{2\pi k_0 R_b^2}{\sqrt{3}ws_\nu^2}\frac{N}{\alpha^2} \tag{3}$$

where $s_\nu$ is the $\nu$-th solution of $J_0(x) = 0$ and $\alpha = (R_b - R_a)/R_a$ is the asymmetry parameter. For the current case ($R_a = 8\,$mm and $f = 15.0\,$GHz), the second solution ($\nu = 2$) corresponds to the object's monopole resonance, marked as $TM_{02}$ mode. Clearly, $Q$ factor is proportional to $N$ and inversely proportional to $\alpha^2$. This is a universal behavior for the $Q$ factor of a quasi-BIC as a function of the asymmetry parameter[44]. Figure 4c shows the relation between $Q$ and $\alpha$ for different $N$ (red, yellow, and blue curves refer to the cases of $N = 2$, 3, and 5, respectively). The analytical results are calculated using Eq. (3), and the numerical ones are obtained from COMSOL. Both analytical and numerical results agree with each other, with the slight deviation seen for large $N$ caused by the theoretical analysis's approximations. In particular, the condition $\alpha = 0$ leads to an infinite $Q$, which proves the existence of the BICs theoretically. For $\alpha$ deviating from zero, the quasi-BICs may occur with a finite and high $Q$ factor. For a fixed $\alpha \neq 0$, the $Q$ factor may be increased mainly by adding objects with $R_a$, leading to further enhancement of EM fields confined inside the objects (see Fig. 4d). This result is further confirmed by eigenmode analysis (see Supplementary Note 1 and Supplementary Fig. 3). This finding provides a feasible way to enhance optical structures' dielectric sensitivity to external perturbations by simply incorporating multiple objects of the same size.

**The frequency response of BICs.** It is known that the frequency response of a ZIM is usually narrowband, i.e., ZIM-based devices only work at a single frequency. In spite of this, the BIC discussed above can still be found in the spectrum because the BIC with an extreme $Q$ factor has a narrower bandwidth response

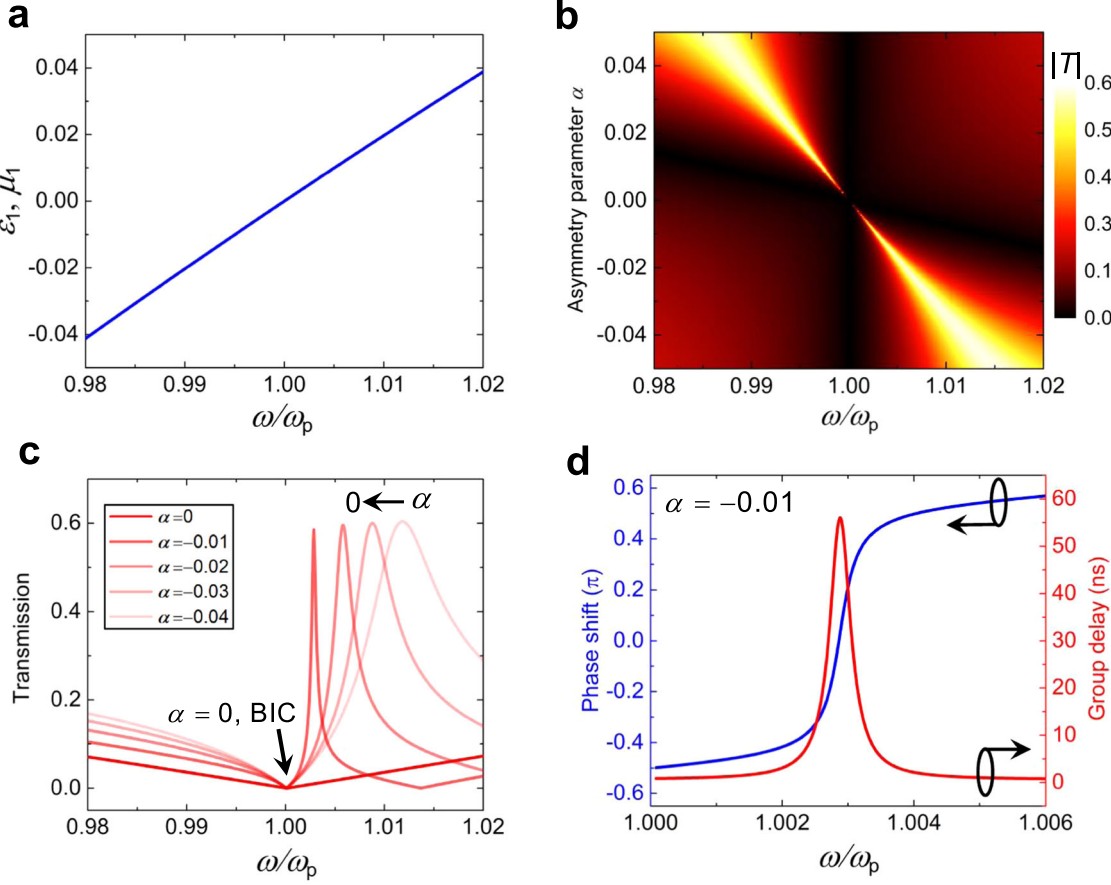

**Fig. 5 BIC frequency response in a dispersive ZIM with two cylindrical objects. a** Drude model for ZIM media. **b** Transmission coefficient as a function of frequency and $\alpha$ for fixed $R_1 = 8$ mm and $\varepsilon_d = 4.82$. **c** Transmission coefficient as a function of frequency: quasi-BIC turns to BIC as $\alpha \to 0$. **d** The phase shift in the transmitted light and group delay due to quasi-BIC.

compared to that of the ZIMs. To illustrate this point, we assume the ZIM to be a Drude-like material with parameters $\varepsilon_1 = \mu_1 = 1 - \omega_p^2/[\omega(\omega + i\gamma)]$, where $\omega_p$ is plasma frequency and $\gamma$ is the damping. To make the discussion simpler, we set $\omega_p = 2\pi \times 15 \times 10^9$ rad/s and ignore the damping, i.e., set $\gamma = 0$. Figure 5a shows the corresponding dispersion relationship, in which for $0.98 \le \omega/\omega_p \le 1.02$, the material features a near-zero profile, ranging from $-0.02$ to $0.02$. With this dispersion, Fig. 5b shows the calculated transmission coefficient as a function of $\omega$ and $\alpha$ for $N = 2$ case. We can clearly observe BIC features similar to those in Fig. 2a. For a fixed $\alpha$, a transmission resonance of quasi-BIC emerges in the spectrum close to $\omega/\omega_p = 1$, and as $\alpha \to 0$, it becomes shaper and eventually disappears (see Fig. 5c). These results further confirm that the case $\alpha = 0$ is associated with an ideal BIC. This form of BIC actually provides a mechanism to slow the speed of light, thus enhancing the inter-action between light and matter. Fig. 5d shows the calculated transmission phase shift $\phi(\omega)$ (blue curve) and the delay time (the red curve) by using the relation[45] $\tau = \partial\phi(\omega)/\partial\omega$ for a quasi-BIC mode with $\alpha = -0.01$. A noticeable delay is seen at $\omega/\omega_p \approx 1$. As $\alpha \to 0$, the phase shift gets steeper and steeper, and when $\alpha = 0$ corresponding to an ideal BIC, the phase shift abruptly change at $\omega/\omega_p \approx 1$, just like a step function, which means an infinite delay time, at least theoretically.

**Practical design and demonstration of ZIM-based BICs using a structural waveguide.** Realistic ZIMs are lossy, and this fact may erase almost all ZIM-based effects. However, the loss of dielectrics

or metals can be negligible at microwave frequencies. There are also some methods to implement a lossless ZIM, such as the Dirac-cone-like photonic crystals[46]. Alternatively, effective ZIMs with matched impedance may be implemented in waveguide systems[36], consisting of two rectangular waveguides (port 1 and port 2) and a waveguide junction. By operating at a working frequency close to the cutoff frequency of each guided mode (e.g., $TE_{10}$ mode), the effective permittivity of the waveguide junction is that of an epsilon-near-zero (ENZ) medium, and its effective permeability could be tuned by doping the waveguide junction with suitable dielectric rods, thus realizing an effective ZIM with matched impedance (see Fig. 6a). In our design, the height of the waveguide, i.e., the separation of two parallel metallic plates in the z-axis, is $H = \lambda_0/2$, and the operating frequency is $f_0 = 2.5$ GHz, corresponding to the cutoff frequency of $TE_{10}$ mode. The cross-section of the waveguide junction is a square with a side $L = 240$ mm, and the width of the input (output) waveguide port is $W = 42$ mm. A dielectric material (Teflon, $\varepsilon_t = 2$, blue areas in Fig. 6a) is used to fill in the input and output waveguides, such that $TE_{10}$ mode above the cutoff frequency is supported. A silicon rod is inserted in the waveguide junction to tune the effective perme-ability, and 16 metallic wires with a diameter of 3.18 mm are placed (equally spaced) around it to form a circle with radius 26.8 mm, which allows it to avoid excitation of other guided modes. Since the setup is aimed at demonstrating quasi-BIC in this ZIM environment, at least additional two objects (e.g., two rods) are required. To this aim, two identical circles (each one made of 16 metallic wires) are also involved in the waveguide junction (see Fig. 6a).

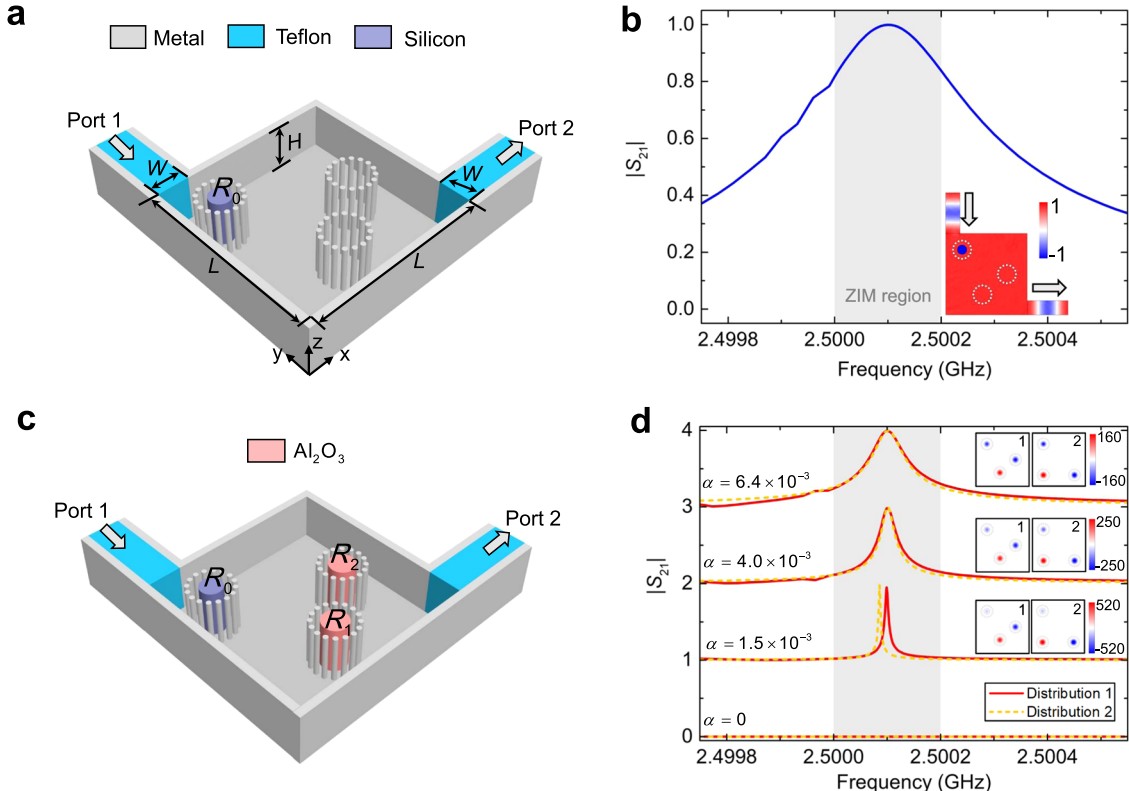

**Fig. 6 Experimental design for ZIM-induced BIC in a metallic rectangular waveguide system. a** Schematic diagram of a metallic rectangular waveguide system with ZIM. The upper parallel metallic plate is hidden to show the internal structure clearly. **b** Numerically calculated transmission spectrum $|S_{21}|$ of the designed waveguide system with ZIM. The grey region shows the parameter region of $\varepsilon_{eff} \approx \mu_{eff} \approx 0$. The inset shows the magnetic field in the middle plane ($z = H/2$) at the resonant transmission. **c** Schematic diagram of the metallic rectangular waveguide system for observing BIC, where two dielectric rods (aluminium oxide) are added in the designed waveguide system with ZIM. **d** Transmission spectrum $|S_{21}|$ for the aluminium oxide rods with different $\alpha$. The red solid and yellow dashed curves denote the two different distributions of aluminium oxide rods. The insets show the corresponding magnetic field patterns in the waveguide junction (square area) with doped rods. Simulation parameters of $R_1$ and $R_2$: $\alpha = 6.4 \times 10^{-3}$ ($R_1 = 16.176$ mm and $R_2 = 16.280$ mm), $\alpha = 4.0 \times 10^{-3}$ ($R_1 = 16.195$ mm and $R_2 = 16.260$ mm), $\alpha = 1.5 \times 10^{-3}$ ($R_1 = 16.215$ mm and $R_2 = 16.240$ mm), $\alpha = 0$ ($R_1 = R_2 = 16.227$ mm). In all simulations, the metals are the perfect electric conductor (PEC) and the material parameters are $\varepsilon_t = 2$, $\varepsilon_{Si} = 11.7$, $\varepsilon_d = 9$.

Based on such a waveguide configuration, we numerically obtain the transmission spectrum ($|S_{21}|$), which is in Fig. 6b, where the radius of the silicon rod is set to $R_0 = 14.087$ mm. High transmission, beyond 80%, is found in a narrow band (grey region) from 2.5 to 2.5002 GHz, caused by the near-zero index region ($\varepsilon_{eff} \approx \mu_{eff} \approx 0$) (see the analytical results in Supplementary Note 4 and Supplementary Fig. 10). In particular, a unity transmission occurs at 2.5001 GHz, which means that an effective ZIM with matched impedance ($\varepsilon_{eff} = \mu_{eff} \to 0$) is obtained as indicated by Supplementary Fig. 10. Also, the inset in Fig. 6b shows the corresponding magnetic field pattern in the waveguide at 2.5001 GHz. Total transmission is observed, and the constant phase distribution appears in the waveguide junction, also implying that an effective ZIM with matched impedance is achieved. Then, we add two dielectric rods (aluminium oxide) with $\varepsilon_d = 9$ into the waveguide, as shown in Fig. 6c (distribution 1). By changing the asymmetry $\alpha$ of the two aluminium oxide rods, the transmission spectra shown in Fig. 6d are obtained (red curves). For $\alpha \to 0$, the transmission peak occurring at 2.5001 GHz becomes narrower, which means that a quasi-BIC is obtained with a higher $Q$ factor. The corresponding magnetic field distributions in the waveguide junction are shown in the insets of Fig. 6d, where the corresponding field distribution in two aluminium oxide rods are out of phase, and their amplitudes are stronger for $\alpha \to 0$. Next, we change the locations of $R_1$ and $R_2$ (distribution 2). As can be seen from the yellow dashed curves,

the ultrasharp spectra are almost the same as their counterparts coming from distribution 1. This confirms that the ZIM-based BIC is not affected by the structural disorder. These results are consistent with the ideal ones reported in Fig. 2. For $\alpha = 0$, the resonant peak disappears in the transmission spectrum, indicating the presence of an ideal BIC.

**Discussion**

In conclusion, upon exploiting the static-like environment of ZIM in 2D waveguide, we have theoretically proposed and numerically demonstrated the existence of geometric-symmetry-free BICs. Those BIC modes are determined by the nontrivial zero of total magnetic flux in the objects in the ZIM host, and exist independently on the shape, position, and filling material of the objects, which are unique features compared with previous results. Remarkably, higher-order BICs may also easily achieved by doping the ZIM host with more objects. More precisely, $N$ identical objects have been shown to induce $(N-1)$-th order BICs, and this result provides a feasible and straightforward way to manipulate the $Q$ factor of quasi-BIC mode. Although the existence of geometric-symmetry-free BICs has been revealed using ZIMs with matched impedance, similar results may also be obtained in an ENZ-based host (see Supplementary Note 5 and Supplementary Fig. 11). In principle, the proposed ZIM-based BICs may be also extended to higher frequency, such as optical regime, considering the experimental advance of ZIM from

microwaves[46] to communication wavelengths[47,48]. Still, considerable implementation complexity, such as lossless ZIM, structural optimization for both ZIM and dielectric obstacles, will be challenging. Our findings offer a way to implement an optical BIC mode without stringent requirements on the system's geometry and enable flexible manipulation of robust, sharp resonances. We foresee promising applications in ultra-fast optical switches, filters, and sensors.

## Methods

**Numerical simulations**. The data in Figs. 2b, 4c, and 5b–d, and the field patterns in Figs. 2c, d, 3b, 4b–d, and the insets of Fig. 6b, d were obtained using the finite element solver COMSOL Multiphysics. For eigenmode analysis, in simulations, we studied a case of a ZIM background containing two same objects of $\varepsilon_d = 4.82$ instead. A large ZIM area under a scattering boundary condition replaces the infinite ZIM. We obtained two eigenfrequencies and corresponding filed pattern, as shown in Fig. 3b. Similar procedures were applied to the case of three cylinders with $\varepsilon_d = 4.82$ and $R_1 = R_2 = R_3 = 8$ mm embedded in the ZIM environment. Based on numerical calculations, we can easily get each eigenmode's eigenfrequency and its corresponding field pattern, as shown in Fig. 4b. In order to save memory, PEC boundary conditions are used to represent the metal during the simulation of Fig. 6b, d.

## Data availability

The data that support the findings of this study are available from the corresponding author upon reasonable request.

## Code availability

The code used for the analyses will be made available upon e-mail request to the corresponding author.

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

## Acknowledgements

This work was supported by The National Natural Science Foundation of China (grant Nos. 11974010, 11904169, 11774252 and 92050104); the Natural Science Foundation of Jiangsu Province (BK20190383); the project funded by the China Post-doctoral Science Foundation (grant Nos. 2018T110540 and 2020M681576); and the Priority Academic Program Development (PAPD) of Jiangsu Higher Education Institutions.

## Author contributions

Y.X. and L.G. conceived the idea. Q.Z., Y.F., L.H., and Y.X. performed the theoretical calculation and numerical simulations. L.G., Q.W., and A.M. helped with the theoretical interpretation. Y.X., Y.F., and L.G. supervised the project. All authors discussed the results and prepared the manuscript.

## Competing interests

The authors declare no competing interests.
