## [Peer Review File · Nature Communications]

Reviewers' Comments:

Reviewer #1:

Remarks to the Author:

The manuscript is well-written and addresses an important problem of bound states in the continuum (BIC), namely instability of symmetry protected BIC in real systems based on geometric symmetry due to the fabrication imperfections. The suggested is based on the structures with zero-index material possessing physical symmetries (parity and time-reversal) rather than geometrical which allows these structures to support BIC. The idea is novel, and the topic might be interesting for the journal community. I have no doubts about the correctness of the results but unfortunately, I cannot recommend the manuscript for acceptance because of the following reasons:

Major criticisms:

- 1) As I can see from the results, although the proposed method suggests spatial symmetry-free BIC, it bounds us with the exact permittivity of voids which is even harder to control during the fabrication. At the same time, conventional symmetry-protected BIC exists at any permittivity. Therefore, it is difficult to state on practical advantages of such BIC without experimental confirmation.
- 2) Moreover, the proposed BIC still requires the geometrical symmetry of void sizes (the asymmetry parameter $\alpha = 0$). That makes unclear the benefits of the author's ZIM-based design comparing with the symmetry-protected BIC. If the obstacles are different, it is possible to find the permittivities (for each obstacle) resulting in the appearance of BIC? If yes, it is also would be reasonable to show this theoretically and experimentally.
- 3) Could the Authors demonstrate the proposed effect for different shapes of the obstacles but not only cylindrical ones?
- 4) How does the Q-factor of the mode changes if the permittivity deviates from the optimal value?

Minor issues:

- 5) The geometry of the problem is unclear and it takes much time to set the relation between Fig. 1 and Fig. 2c,d, for example.
- 6) There are a lot of puzzles the readers need to solve to understand what the Authors mean. For example, to understand what is H1 constant in eq. (1) it is necessary to read Ref.[30] or how the wave does propagate in Fig. 2c,d, etc.
- 7) At the beginning of Sec. "Models and Theory" there is a typo "with even and odd symmetry relative to the x-axis." It should be "y-axis".
- 8) It is not clear why the Authors plot either $\text{Re}(Hz)$ or $\text{Im}(Hz)$ in Fig.2c,d.
- 9) It is reasonable to cite this paper [Physical Review B 78.7 (2008): 075105.] along with [1,13,18]
- 10) It follows from Eq. (3) the $Q \sim 1/\alpha^2$. A similar result was obtained in [Physical review letters 121.19 (2018): 193903]. Thus, it is reasonable to cite this paper too.
- 11) Fig. 5a is trivial.
- 12) Authors appeal several times to optical systems but it seems that such an approach will work well only for the radiofrequency range.

Reviewer #2:

Remarks to the Author:

This paper investigates a mechanism which leads to symmetry protected bound states in continuum (BICs) which are insensitive to the geometry of the system, i.e., BICs that are robust to any geometrical imperfections. This work takes advantage of static-like field distribution in zero-index metamaterials (ZIMs) where the effective permeability of a 2D ZIM can be modified by microscopic dielectric inclusions while its effective permittivity remains unchanged. In another word, the composite ZIM and dielectric inclusions acts as a material with a homogeneous refractive index. This means any number of dielectric objects can be immersed in the ZIM at arbitrary locations, but still the homogeneous refractive index is unchanged. This fact sets the basis in this work to provide a system that demonstrates robust BICs without any geometrical symmetry. However, the BIC is still protected by physical symmetry (properties of the objects immersed in

the ZIM not their location).

The results claimed in this work are backed by analytical and numerical methods i.e., it has been shown that the BICs are robust against geometrical modifications by designing different shapes of dielectric inclusions and placing them in arbitrary locations inside the ZIM. Also, in the last section a practical design is suggested to realize these BICs. I found the work and its context suitable for publication, however there is still one confusion which needs authors' elaboration:

As it is stated in the draft, "the existence of robust geometric-symmetry-free BICs, results from a nontrivial zero value of the total effective magnetic flux". But for the case of $a=0$ (exactly zero asymmetry parameter) the two resonances peaks of the dielectric inclusions coincide, therefore, there are two-fold degenerate states in the two objects case. In this situation one state has infinite flux and the other has zero flux. But then the total flux is not zero which seems to be in contradiction to the main claim of the work. Also, what is the intuitive physical origin of this conjecture for $a=0$, i.e., the existence of two states, dark and bright states?

Reviewer #3:

Remarks to the Author:

I read the article "Geometry symmetry-free and Higher-order Optical Bound States in the Continuum", and it deals with the emergence of non-radiative states in zero-parameter materials. The authors write that their "results are very different from those reported previously for ZIM-based optical systems [35, 36]". I am afraid I disagree: the physical mechanism is exactly the same. In fact, as written in the supplementary materials the BIC occurs when the transmission coefficient has a valley, which corresponds to a zero of the (cylindrical) Bessel function J_0 . This is exactly the same condition derived in previous works. The BIC predicted by the current article is precisely the same that was discussed in the above mentioned references, except that here the geometry is two-dimensional, rather than three-dimensional.

Finally, I note that the fact that the BICs are "geometry symmetry-free" was discussed (for the three-dimensional case) here:

I. Liberal and N. Engheta, "Nonradiating and radiating modes excited by quantum emitters in open epsilon near-zero cavities," *Sci. Adv.* 2, e1600987, 10 (2016).

which also considered configurations with multiple cavities surrounded by a ENZ material. Due to the above reasons, in my opinion the article has very little novelty and is more suitable to a specialized journal.

Response to the reviewer's comments

We thank the reviewers for their comments and suggestions, which are helpful to improve our manuscript further. In the resubmitted manuscript, we have made revisions accordingly, and all the changes are marked in red. Below is our response to the reviewers' comments.

Reviewer #1

The manuscript is well-written and addresses an important problem of bound states in the continuum (BIC), namely instability of symmetry protected BIC in real systems based on geometric symmetry due to the fabrication imperfections. The suggested is based on the structures with zero-index material possessing physical symmetries (parity and time-reversal) rather than geometrical which allows these structures to support BIC. The idea is novel, and the topic might be interesting for the journal community. I have no doubts about the correctness of the results but unfortunately, I cannot recommend the manuscript for acceptance because of the following reasons:

Reply: We thanks for the reviewer's positive comments, and also appreciate these valuable suggestions that are helpful to further improve the quality of our work. According to the reviewer's suggestions/ comments, we have made more efforts to further clarify our findings and make them more convincing.

Major criticisms:

1. As I can see from the results, although the proposed method suggests spatial symmetry-free BIC, it bounds us with the exact permittivity of voids which is even harder to control during the fabrication. At the same time, conventional symmetry-protected BIC exists at any permittivity. Therefore, it is difficult to state on practical advantages of such BIC without experimental confirmation.

Reply: We thanks the reviewer for pointing out the confusing point. Here, we would like to emphasize that the proposed symmetry-free BIC happens at the condition of nontrivial zeros of the total magnetic flux in all the dielectric voids. This specific corresponds to the root of $J_0(k_d R_i)$ in a simple case, i.e., two identical dielectric voids with circle shape. Thus, the proposed symmetry-free BIC can also happen at any permittivity, but the working frequency and/or the size of voids should be accordingly tuned. In our study (Fig. 2 - Fig. 4), for the sake of convenience, we fix both the working frequency at 15.0 GHz by considering the narrow bandwidth of ZIM and the size of voids, but only change the permittivity to observe the BICs and reveal the physical mechanism. As a result, it seems that the BICs only happen at the same permittivity of 4.82, which leads to the misunderstanding.

We calculate the transmission spectrum mapping versus frequency and asymmetric parameter for the voids with different permittivity to resolve this misunderstanding, as shown in Fig.R1. Note that BIC can be identified easily by checking the vanishing linewidth. As seen from Fig. R1a-c, the BIC is red-shifted and preserved with the increase of permittivity. The working frequencies are slightly shifted accordingly, but still located in the frequency window of ZIM, whose bandwidth is about 4%, as displayed in Fig. 5a.

Similar to the conventional symmetry-protected BIC, the ZIM-based BIC also exists at any permittivity. Therefore, dielectric voids' permittivity could be an arbitrary value, as long as the working frequency of the proposed BICs is located in the ZIM window. Besides, by tuning the working frequency in the ZIM window and the size of voids, the permittivity is changed to $\epsilon_d=9$, corresponding to the permittivity of Al_2O_3 in microwaves. Detailed experimental design for ZIM induced BIC in a metallic waveguide system can be found in Fig.6. Prof Nader Engheta from UPenn has demonstrated the feasibility of implementing ZIM in such a waveguide system in Ref [36] (Science 355, 1058 (2017)), which constitutes the critical element of our design.

Fig. R1. Transmission as a function of frequency and asymmetry parameter at different permittivity of voids. The model is the same as Fig. 2a in the main text.
a $\epsilon_d = 4.7$; **b** $\epsilon_d = 4.82$; **c** $\epsilon_d = 5$.

In the revised manuscript, we have added some words to explain clearly this confusing point, reading as.

"In the analysis, we fix the working frequency at 15.0 GHz by considering a narrow bandwidth of the ZIM and also keep the size of one object ($R_1=8$ mm), only change the permittivity of objects and the asymmetry parameter to observe the proposed BICs conveniently."

And

"..., it is noted that similar to the conventional symmetry-protected BIC, the proposed BIC can exist at any permittivity for the objects with a fixed size, as long as its shifted working frequency is still located in the frequency window of ZIM (Supplementary Fig. 1)."

Therefore, similar to the conventional symmetry-protected BIC, the ZIM-based BIC also exists at any permittivity of dielectric voids, as long as the working frequency of the proposed BICs is located in the ZIM window.

2. Moreover, the proposed BIC still requires the geometrical symmetry of void sizes (the asymmetry parameter $\alpha = 0$). That makes unclear the benefits of the author's ZIM-based design comparing with the symmetry-protected BIC. If the obstacles are different, it is possible to find the permittivities (for each obstacle) resulting in BIC appearance? If yes, it is also would be reasonable to show this theoretically and experimentally.

Reply: We think the reviewer has raised a constructive comment. In fact, the proposed

ZIM-based BICs can be implemented by the objects with arbitrary external shapes and any permittivity because the proposed BICs result from the nontrivial zeros of the total magnetic flux in the objects, which offers excellent flexibility for the material parameter and geometric shape of all the objects.

To clearly demonstrate this unique property, we consider a more general model, i.e., a rectangular object with $\epsilon_d = 9$ and a cylinder with $\epsilon_d = 4.82$, as shown Fig. R2a. In this case, we cannot define the asymmetry parameter for two objects. The rectangular object with a fixed size of $a=6$ mm and $b=4.01$ mm support a monopole mode at 15.0 GHz. Figure R2b shows analytical transmission spectra as a function of the working frequency and the radius R of a cylinder, and we assume ZIM is non-dispersive. With the increase of R , the monopole mode of the cylinder redshifts gradually. When the radius is increased to $R=8.0$ mm, the monopole modes of two objects degenerate, and a vanishing linewidth is observed at 15.0 GHz. We perform eigenmode analysis for this condition, and a radiation mode and BIC mode are found in Fig. R2c. In particular, these two objects' magnetic fluxes have opposite directions, leading to nontrivial zero of the total magnetic flux, i.e., the BIC mode.

Furthermore, Fig. R2d shows transmission spectrum with $R=8.1$ mm, and analytical results agree well with simulations. The first dip can be correlated to the cylinder's monopole mode, which corresponds to zero of the Bessel function $J_0(k_c R) = 0$. The second dip results from the monopole mode of the rectangular object, which correspond to $k_r^2 - (\pi/a)^2 - (\pi/b)^2 = 0$. All these results have been added to supplementary information.

More details can be seen in Supplementary Note 2 and Supplementary Fig. 8. Therefore, the proposed ZIM-based BICs can exist independent of the external shape, material parameter of all the objects, as they are protected by physical symmetry but not geometrical symmetry.

Fig. R2. Geometric-shape free and materials-independent BIC in the frequency domain. **a**, A rectangular object with permittivity $\epsilon_d = 9$ and a cylinder with permittivity $\epsilon_d = 4.82$ embedded in non-dispersive ZIM. The length and width of the rectangle are $a=6$ mm and $b=4.01$ mm, respectively. The radius of the cylinder is R . The other parameters are the same as Fig. 2a. **b**,

Transmission as a function of frequency and the radius of the cylinder. An ideal BIC occurs at $f=15$ GHz and $R = 8$ mm (the white dash circle). (c) Eigenmode analysis for a rectangular object and a cylindrical object embedded in the ZIM environment. Filling contours are an out-of-plane magnetic field. A symmetric mode and an antisymmetric mode become degenerated at 15 GHz. The rectangular object shares the same size as that in **a**, and $R=8$ mm. **d**, Transmission spectrum for $R=8.1$ mm. Insets show the magnetic field patterns at the first dip, peak, and second dip.

In the revised manuscript, we added some discussion to explain this confusing point, reading as clearly.

"Moreover, as the BICs result from the nontrivial zeros of the total magnetic flux in the objects, it offers great flexibility for the material parameter and geometric shape of all the objects in ZIM (see Fig. 1c). To demonstrate these features, we numerically studied several cases of dielectric objects with square, triangle, arbitrary external shapes and different filling materials. In all cases, similar BICs are found, always coming from the nontrivial zero of total flux in the two objects with the anti-symmetry-like mode state, which are displayed respectively in Supplementary Figs. 4-7. In particular, even in a more general model of two objects consisting of a rectangular object with $\epsilon_d = 9$ and a cylinder with $\epsilon_d = 4.82$, the proposed BIC is revealed analytically and numerically (see Supplementary Note 2 and Supplementary Fig. 8), which well confirms the existence of the proposed BIC independent of the material parameter and geometric shape of these objects in ZIM."

3. Could the Authors demonstrate the proposed effect for different shapes of the obstacles but not only cylindrical ones?

Reply: We thank the reviewer for bringing this to our attention. The effect is applicable to objects of arbitrary shape. In fact, in the previous manuscript, several examples have been displayed to demonstrate this effect (Supplementary Fig. 4-7). We also consider more general cases in Supplementary Fig. 8, as discussed in Reply to comment 2.

4. How does the Q-factor of the mode changes if the permittivity deviates from the optimal value?

Reply: As explained in Reply 1, actually, the BIC can exist in any permittivity of the objects, and there is no optimal value for the permittivity. With the change of the objects' permittivity, the BIC with infinite Q factor only is red-shifted or blue-shifted (see Fig. R1), but the working frequencies are still located in the frequency window of ZIM. To quantify the Q factor of a BIC mode (e.g., a quasi-BIC) with the change of the permittivity, we plot the relationship between the Q factor of the quasi-BIC and the permittivity of objects in Fig. R3, in which we consider a case of the permittivity slightly deviating from 4.82, ranging from 4.7 to 5.0. As seen, the Q factor is almost the same. In fact, this point is also found from Eq. (1) or Eq. (3) which does not contain ϵ_d .

Fig. R3. Q factor of a quasi-BIC vs the permittivity of cylinders. The model is the same as Fig. 1b in the main text, and $\alpha=-0.01$. The blue solid line is for analytical results based on Eq. (1), and the red balls show the numerical results marking the Q-factor of Fig. R1a-c with $\alpha=-0.01$.

Minor issues:

1. The geometry of the problem is unclear and it takes much time to set the relation between Fig. 1 and Fig. 2c, d, for example.

Reply: Thanks for the reviewer's comment. To clearly show the main idea of ZIM-based BICs, we re-plot Fig.1. The new Fig. 1b is the case of two cylinders in ZIM, which can directly relate to the structure of Fig. 2c, d. To further reveal more features of the ZIM-based BICs, we added two more schematic diagrams (Fig. 1c, d) to describe geometric-symmetry-free BIC and higher-order BICs.

In the revision, more words are added to explain new Fig. 1, reading as,

"However, when one of the objects is spatially shifted (the circle with dashed boundary in Fig. 1a), the reflection symmetry of the system is lost, leading to the disappearance of the robust BIC mode. In the following, we will show that such BIC mode could be preserved by introducing the background medium of ZIM for these two objects (see Fig. 1b), even if the external shapes, materials and number of these objects are arbitrary changed (see Fig. 1c, d)."

2. There are a lot of puzzles the readers need to solve to understand what the Authors mean. For example, to understand what is H_1 constant in eq. (1) it is necessary to read Ref.[30] or how the wave does propagate in Fig. 2c,d, etc.

Reply: Thanks for the reviewer's comments. Indeed, some of the necessary points were not clearly explained. According to your suggestions, we make the following four significant changes in the Model and Theory part:

(1) We add a sentence to explain the meaning of each parameter in Eq. (1), which read as, " H_1 is a local constant of the magnetic field in the ZIM depending on the incident wave, $\omega = 2\pi f$ is the angular frequency of the incident wave, and f is the working frequency".

(2) We also find that " k_2 " in Eq. (2) is not consistent with the supplementary material. So in the revision, the " k_2 " in Eq. (2), is changed into " k_d ".

(3) In Eq. (2), an additional explanation is added, which read as,

" $k_d = \sqrt{\epsilon_d} \omega/c$ is the wave vector in the objects, c is the speed of light in vacuum".

(4) The description of the incident wave is added, which reads as,

"We consider a TM wave normally incident from the left port."

3. At the beginning of Sec. "Models and Theory" there is a typo "with even and odd symmetry relative to the x-axis." It should be "y-axis".

Reply: Thanks for the reviewer's careful review. We define the modes' even and odd symmetry by regarding the dash-dot line along x-direction as the symmetry axis. To make it easier to understand, we added more words to explain it, which reads as clearly,

"the waveguide supports two continuous waves (guide modes), with even and odd symmetry with respect to the axis of symmetry of the waveguide (the dash-dot line)."

4. It is not clear why the Authors plot either $\text{Re}(H_z)$ or $\text{Im}(H_z)$ in Fig.2c, d.

Reply: We thanks the reviewer to point out this confusing point. In fact, there is no essential difference between $\text{Re}(H_z)$ and $\text{Im}(H_z)$, with only the phase difference of $\pi/2$ between them. Generally, it's common to use $\text{Re}(H_z)$ to represent the field distribution. In our simulations, the amplitude of the incident wave is set to be unity. When the TM incident wave impinges the left boundary of ZIM, the real part of the magnetic field is 1 A/m, due to zero initial phase. At the condition of total reflection (e.g., N/M point in Fig. 2c, d), the $\text{Re}(H_z)$ distribution inside cylinder objects is too weak (see Fig. R4 b, d for total reflection at N point). If we show the $\text{Re}(H_z)$ distribution inside cylinders, it will bring the misunderstanding of weak field intensity. To better show the field enhancement inside cylinders, we therefore plotted the $\text{Im}(H_z)$ distribution (see Fig. R4 a, c) instead for the total reflection cases corresponding to the N/M point.

Fig. R4. Details of magnetic field distribution for total reflection at N point. **a** and **b** represent the $\text{Im}(H_z)$ and $\text{Re}(H_z)$ at total reflection, respectively. The ZIM regions are marked by yellow dashed boxes. **c** and **d** represent the field patterns of **a** and **b** in ZIM regions with larger color bars, respectively. The imaginary part is several orders of magnitude larger than the real part.

In the revised manuscript, we added some words to explain this point, which read as clearly, "In simulations, the real part of the magnetic field is 1 A m^{-1} for incident wave arriving at the left boundary of ZIM. The imaginary (real) part of magnetic field in the objects is dominant

for the case of total reflection (total transmission). To clearly reveal the field enhancement and the phase relationship in the objects, the corresponding imaginary/real part of the magnetic field in the ZIM region of Fig. 1b is shown in Fig. 2c and Fig. 2d."

5. It is reasonable to cite this paper [Physical Review B 78.7 (2008): 075105.] along with [1,13,18]

Reply: Thanks for the reviewer's suggestion. The mentioned paper is excellent work discussing the symmetry-protected BIC with discrete frequencies belonging to the continuum for different arrangements of the photonic crystal defects; it has been cited in the revision.

6. It follows from Eq. (3) the $Q \sim 1/\alpha^2$. A similar result was obtained in [Physical review letters 121.19 (2018): 193903]. Thus, it is reasonable to cite this paper too.

Reply: Thanks for the reviewer's suggestion. This paper has been cited in the revision. Also, we have added some words to discuss Eq. (3), which reads as,

"This is a universal behavior for the Q factor of a quasi-BIC as a function of the asymmetry parameter [43]."

7. Fig. 5a is trivial.

Reply: It seems Fig. 5a is trivial, but it is much better to present, considering the reason that it can directly show that the bandwidth of a BIC mode is much narrower than that of the dispersive ZIM. We hope the reviewer's understanding.

8. Authors appeal several times to optical systems but it seems that such an approach will work well only for the radiofrequency range.

Reply: Indeed, the proposed ZIM-based BICs is easily implemented in the microwave range. Considering the advances of ZIM at around 1550 nm [Phys. Rev. Lett. 121, 263901 (2018)] and the corresponding experimental results [Nano Lett. 21, 914-920 (2020)], the key results present by this work could, at least in theory, be observed in optical regimes. In these two works, a quasi-3D photonic crystal of a dielectric slab drilled with periodic array of air holes was studied, which can enable a lossless effective ZIM at a specific frequency. Of course, this proposal's realization in optical systems might be challenging due to loss and fabrications.

In the revision, we also added some words to discuss this problem in the conclusion part, which reads as,

"In principle, the proposed ZIM-based BICs may be also extended to higher frequency, such as optical regime, considering the experimental advance of ZIM from microwaves⁴⁵ to communication wavelengths^{46, 47}. Still, considerable implementation complexity, such as lossless ZIM, structural optimization for both ZIM and dielectric obstacles, will be challenging."

Reviewer #2

This paper investigates a mechanism which leads to symmetry protected bound states in continuum (BICs) which are insensitive to the geometry of the system, i.e., BICs that are robust to any geometrical imperfections. This work takes advantage of static-like field distribution in zero-index metamaterials (ZIMs) where the effective permeability of a 2D ZIM can be modified by microscopic dielectric inclusions while its effective permittivity remains unchanged. In another word, the composite ZIM and dielectric inclusions acts as a material with a homogeneous refractive index. This means any number of dielectric objects can be immersed in the ZIM at arbitrary locations, but still the homogeneous refractive index is unchanged. This fact sets the basis in this work to provide a system that demonstrates robust BICs without any geometrical symmetry. However, the BIC is still protected by physical symmetry (properties of the objects immersed in the ZIM not their location). The results claimed in this work are backed by analytical and numerical methods i.e., it has been shown that the BICs are robust against geometrical modifications by designing different shapes of dielectric inclusions and placing them in arbitrary locations inside the ZIM. Also, in the last section a practical design is suggested to realize these BICs. I found the work and its context suitable for publication, however there is still one confusion which needs authors' elaboration:

Reply: We thank the reviewer for the highly remarkable comments.

As it is stated in the draft, "the existence of robust geometric-symmetry-free BICs, results from a nontrivial zero value of the total effective magnetic flux". But for the case of $a=0$ (exactly zero asymmetry parameter) the two resonances peaks of the dielectric inclusions coincide, therefore, there are two-fold degenerate states in the two objects case. In this situation one state has infinite flux and the other has zero flux. But then the total flux is not zero which seems to be in contradiction to the main claim of the work. Also, what is the intuitive physical origin of this conjecture for $a=0$, i.e., the existence of two states, dark and bright states?

Reply: The reviewer has raised a good question. For the physical origin, the intuitive picture is the interference of two fields inside the ZIM background that radiate from two monopole modes in the objects. They support radiative monopole modes with an identical resonant frequency in two identical objects with zero asymmetry parameters. They can easily couple with each other and produce a symmetry state and an anti-symmetry (asymmetric) state, corresponding to a bright state and dark state, respectively (see Fig. 3b). The anti-symmetry mode with zero total flux corresponds to a BIC mode, and it has a zero linewidth that totally decoupled with external incidence. The expected total transmission from the condition of zero total flux cannot appear, which is accessible for a quasi-BIC mode due to its narrow linewidth. Instead, the symmetry state (bright state) can be coupled with the external incidence, which leads to the total reflection owing to the infinite flux.

In the revision, to clearly explain this confusing point, we added some words to clarify it further, reading as,

"Intuitively, these two-fold degenerate eigenmodes result from the interference of the fields

inside the ZIM background radiated from monopole modes in the two objects, as they have identical resonant frequency and easily couple with each other. The anti-symmetry mode with zero total flux corresponds to a BIC mode (dark mode), which has a zero linewidth and is totally decoupled with external incidence, as reflected by the zero magnetic fields in the ZIM background (see Fig. 3b). As a result, the expected total Transmission from the condition of zero total flux in Eq. (1) cannot appear (but it is accessible for a quasi-BIC mode with narrow linewidth as it can couple with external incidence). Instead, the symmetry state (bright mode) with a non-zero background field (see Fig. 3b) is coupled with the external incidence, which leads to the zero transmission (see Fig. 2a) owing to its infinite total flux."

Reviewer #3

I read the article "Geometry symmetry-free and Higher-order Optical Bound States in the Continuum", and it deals with the emergence of non-radiative states in zero-parameter materials. The authors write that their "results are very different from those reported previously for ZIM-based optical systems [35, 36]". I am afraid I disagree: the physical mechanism is precisely the same. In fact, as written in the supplementary materials, the BIC occurs when the transmission coefficient has a valley, which corresponds to a zero of the (cylindrical) Bessel function J_0 . This is exactly the same condition derived in previous works. The BIC predicted by the current article is precisely the same discussed in the above mentioned references, except that the geometry is two-dimensional, rather than three-dimensional.

Reply: We thank the reviewer for his/her careful reading of our manuscript. We think that the reviewer may have missed some critical points of the manuscript, leading to some misunderstandings. We would like to state that the novelty and the main results of our work are very different from those reported previously in ZIM-based optical systems [35, 36] and the following mentioned work (Sci. Adv. 2, e1600987, 10 (2016)).

Previous works majorly studied ZIM-based cavities, such as core (dielectric)-shell (ZIM) structure or dielectric particles embedded in ZIM. The demonstrated BIC modes (embedded eigenstates) actually correspond to higher-order cavity modes ($n \geq 1$), such as dipole modes, with the behind mechanism that originates mainly from the unique feature of ZIM itself. Due to extreme material parameter, the doped ZIM structure, to some extent, can be regarded as a perfect electric/magnetic conductor. In such ZIM-based cavities with higher-order modes, e.g., dipole mode, can be highly confined due to non-uniform field distributions in the angular direction. However, **it is impossible for the radiation field of monopole mode in a single object owing to its uniform field in the angular direction**, so that there is common sense that the BIC mode of the cavity mode with $n=0$ (embedded eigenstates) could not be realized, which has been also discussed in a recent paper [Laser Photonics Rev. 2018, 12, 1700220].

This challenge is perfectly resolved in our work by embedding multiple objects into ZIM. A more general condition for BICs in such a system is derived. In this work, we mainly focus on the monopole modes ($n=0$). We show that due to the mechanism of nontrivial zero total magnetic flux ($\sum \Phi_i=0$), the BIC corresponding to the cavity mode with $n=0$ can be

achieved. It should be noted that the presented condition $\sum \Phi_i=0$ for BICs in the doped ZIM is more general, which applies to these two different cases: (i) $\Phi_i=0$ in each object; (ii) $\Phi_i \neq 0$ in each object, but $\sum \Phi_i=0$. The former case corresponds to the previous works, where the higher-order cavity modes lead to zero magnetic flux in each object ($\Phi_i=0$). While the other case corresponds to our work, and the monopole-based BICs come from the zero value of total magnetic flux of all objects ($N \geq 2$) (the magnetic flux of each object is not zero, i.e., $\Phi_i \neq 0$). Strictly speaking, our studied case's condition is $\sum \Phi_i=0$ with $\Phi_i \neq 0$, which is defined by *nontrivial* zeros of total magnetic flux, to distinguish it from the *trivial case*: $\sum \Phi_i=0$ with $\Phi_i=0$. To better understand of these two types of BICs, we show the eigenmode analysis for one cylinder with monopole mode and dipole mode resonances (see Fig. R5 a, c), where BIC is only realized by dipole mode resonance, and also for two same cylinders with monopole mode and dipole mode resonances (see Fig. R5 b, d), where BIC is either realized by dipole mode resonance or monopole mode resonance, which correspond to different mechanisms as we explained.

Fig. R5 Eigenmode analysis for a ZIM environment with objects. **a** and **b** the modes are monopole modes. **c** and **d** the modes are dipole modes. For single object embedded in ZIM, monopole mode is radiation mode, but dipole mode is BIC mode. For two objects embedded in ZIM, the coupling between two monopole modes induce a BIC, but two dipole modes are decoupled.

Besides, the reviewer commented that "...the BIC occurs when the transmission coefficient has a valley, which corresponds to a zero of the (cylindrical) Bessel function J_0 . This is exactly the same condition derived in previous works." Indeed, at first glance, our work does look similar to some of the previous works. However, there are essential differences.

To clarify this misunderstanding, we consider two situations: (a) there is only one cylinder object (i.e. $N=1$) inside the ZIM; (b) there are multiple cylinder objects (i.e. $N \geq 2$) inside the ZIM. For $N=1$ case, the zero of the (cylindrical) Bessel function J_0 induces effectively infinite magnetic flux ($\Phi \rightarrow \infty$), leading to a transmission valley. For $N \geq 2$ case, as long as one of the N objects' cavity modes has the zero of Bessel function J_0 , the transmission valley also happens. In this case, we have $\Phi_j \rightarrow \infty$ and $\sum \Phi_i \rightarrow \infty$. These cases have been well explored in previous works, such as Ref. 31 and 32 in the revised manuscript. But it should be noted that such transmission valley does not correspond to a BIC mode because all cavity modes with the zero of Bessel function J_0 are bright modes, which can be excited by external incident wave. In our work, we consider $N \geq 2$ case, and the BIC modes occur at the condition $\sum \Phi_i = 0$ and $\Phi_i \neq 0$. The BIC is presented by the transmission peak rather than the transmission valley. This point is captured from Fig. 2a and 2b. Therein, for quasi-case with $\alpha \neq 0$, the zero of Bessel function J_0 in two different objects leads to two transmission valleys at different positions. The quasi-BIC mode corresponds to transmission peak with finite linewidth, which is ensured by $\sum \Phi_i = 0$ and $\Phi_1 = -\Phi_2 = \text{finite value}$. For the ideal case with $\alpha = 0$, the ideal BIC presented by the transmission peak with zero linewidth, is embedded inside the transmission valley, which is guaranteed by $\sum \Phi_i = 0$ and $\Phi_1 = -\Phi_2 \rightarrow \infty$.

As we reply in Reviewer #2, in our study, the intuitive physical picture for BICs is the interference of two fields inside the ZIM background radiated from two monopole modes in the objects. In two identical objects with zero asymmetry parameters, they support radiative monopole modes with an identical resonant frequency. They can easily couple with each other and produce a symmetry state and an anti-symmetry (asymmetric) state, corresponding to a bright state and a dark state, respectively (see Fig. 3b). The anti-symmetry mode with zero total flux corresponds to a BIC mode, and it has a zero linewidth that totally decoupled with external incidence. The expected total transmission from the condition of zero total flux cannot appear, which is accessible for a quasi-BIC mode due to its narrow linewidth. Instead, the symmetry state (bright state) can be coupled with the external incidence, which leads to the total reflection owing to the infinite flux.

On the other hand, owing to the revealed mechanism of nontrivial zeros of the total magnetic flux in the objects, it offers great flexibility for the material parameter and geometric shape of all the objects ZIM to implement a BIC mode. We demonstrate our design's robustness by investigating different combinations of voids with arbitrary shape and different permittivity (See Supplementary Fig. 4-Fig. 8). This significantly relaxes the requirement of experimental demonstration. The case of cylindrical objects and the BIC condition of $J_0(kR)=0$ is solely a particular example to easily demonstrate our findings. Besides benefiting from the proposed

BIC via monopole modes in multiple objects ($N \geq 2$), we revealed **the higher-order BICs, to our best knowledge, which has not been discussed in the ZIM system**. Therefore, we believe that "results are very different from those reported previously for ZIM-based optical systems".

Finally, I note that the fact that the BICs are "geometry symmetry-free" was discussed (for the three-dimensional case) here:

I. Liberal and N. Engheta, "Nonradiating and radiating modes excited by quantum emitters in open epsilon near-zero cavities," *Sci. Adv.* 2, e1600987, 10 (2016). which also considered configurations with multiple cavities surrounded by a ENZ material.

Due to the above reasons, in my opinion the article has very little novelty and is more suitable to a specialized journal.

Reply: The mentioned paper also deals with the problem of BICs induced by the higher-order modes (dipole mode) in the doped ZIM cavities, where the excitation of non-radiating and radiating modes is controlled by the symmetry of the dipole source with respect to the cavity mode. As we explained previously, the mechanism is inherently different from our case. The non-radiating mode (BIC) is totally trapped in the cavities surrounded by an ENZ material, which is totally decoupled with other cavities (objects) ensured by resonance solution of $J_1(kR)=0$. As a result, the BIC can exist independently of the external boundary of ENZ and the other dielectric bodies within it. However, our studied BIC results from the interaction between the radiating monopole modes of cavities (objects) in ZIM and these monopole modes can couple with each other. The BIC from monopole modes is caused by the nontrivial zeros of the total magnetic flux in all the objects. Therefore, it can exist independently of the material parameter and geometric shape of all the objects in ZIM (also, the external boundary of ZIM could be arbitrary). Therefore, our results with unique novelty are different from the mentioned paper.

In order to clarify the significant differences between previous works and ours, we added several sentences in the introduction part, which read as

"In this work, we introduce the concept of geometric-symmetry-free but physical-symmetry-protected BICs a ZIM host embedded with $N \geq 2$ objects, which support radiative monopole modes with non-zero magnetic flux. Resulting from a nontrivial zero value of the total effective magnetic flux in the N objects, we will demonstrate the existence of robust geometric-symmetry-free BICs, which could be realized regardless of the specific positions, external shapes and material parameters of the objects in the ZIM host. More interestingly, we find that using N doping objects enables higher-order BICs, and derive an analytical formula for the N -dependent Q factor. Specifically, the N objects can produce $(N-1)$ -fold degenerate BICs. These results are very different from those reported previously for ZIM-based optical systems³⁷⁻³⁹, where these BICs (embedded eigenstates) are induced by the non-radiative higher-order cavities modes (e.g., dipole mode) with zero magnetic flux. Our results break the conventional wisdom of a ZIM-based BIC excluded from monopole modes⁴⁰ and pave a new way to study the higher-order BICs and the associated physics."

And in the main text, reading as,

"Thus, the BIC mode discussed above is physically related to the *nontrivial* zeros of the total magnetic flux, i.e., $\sum \Phi_i = 0$ while $\Phi_i \neq 0$, which distinguishes from the common situation of $\sum \Phi_i = 0$ with $\Phi_i = 0$ ⁴⁰. In fact, $\sum \Phi_i = 0$ is general condition for BICs in the doped ZIM system and applies to two different situations, i.e., radiative monopole modes related to $\Phi_i \neq 0$, and higher-order cavities modes caused by zero magnetic flux in each object ($\Phi_i = 0$)."

Reviewers' Comments:

Reviewer #1:

Remarks to the Author:

The Authors answered carefully all my questions and I completely satisfied with the answers. Therefore, I recommend the manuscript for acceptance.

Reviewer #2:

Remarks to the Author:

Authors have addressed my concern in my previous review and I do not have any question at this point.

Reviewer #3:

Remarks to the Author:

I sincerely thank the authors for their detailed replies and clarifications.

I understand the point that the field in each scatterer may indeed have "locally" a monopole symmetry. However, the *global* symmetry of the BIC is certainly not that of a monopole. For example, for $N=2$ scatterers the global symmetry is the one of a dipole, analogous to the previous works I quoted. So, in the end, what is being studied here a particular dipole-type resonator that is physically spread out inside the ZIM material. Note that a single scatterer may also be "conceptually" split into two parts and regard each of the halves as a monopole.

As to the scattering properties, I still do not see the exact difference of this system and other reported previously. Do not they share a Fano-type lineshape near the quasi-BIC state?

Finally, independent of any technical aspects, I cannot agree with sentences in the introduction of the article such as:

"Therefore, how to achieve robust BICs in a system with no exact geometrical symmetry is still an open question." This sentence is completely misleading as the solution proposed here to solve the problem is not new and is very well understood.

Response to the reviewer's comments

We thank the reviewers for their comments and suggestions, which are helpful to improve our manuscript further. In the resubmitted manuscript, we have made revisions accordingly, and all the changes are marked in red. Below is our response to the reviewers' comments.

Reviewer #1

The Authors answered carefully all my questions and I completely satisfied with the answers. Therefore, I recommend the manuscript for acceptance.

Reply: We great thanks for the Reviewer's recommendation of acceptance.

Reviewer #2

Authors have addressed my concern in my previous review and I do not have any question at this point.

Reply: We great thanks for the Reviewer's recommendation of acceptance.

Reviewer #3

I sincerely thank the authors for their detailed replies and clarifications.

Reply: We thank the Reviewer for these comments and suggestions to improve our work. We also hope that the following response can make the Reviewer satisfied.

I understand the point that the field in each scatterer may indeed have "locally" a monopole symmetry. However, the *global* symmetry of the BIC is certainly not that of a monopole. For example, for $N=2$ scatterers the global symmetry is the one of a dipole, analogous to the previous works I quoted. So, in the end, what is being studied here a particular dipole-type resonator that is physically spread out inside the ZIM material. Note that a single scatterer may also be "conceptually" split into two parts and regard each of the halves as a monopole.

Reply: We think the Reviewer has raised an insightful understanding of ZIM-based BIC modes induced by a dipole mode in a single scatterer and monopole modes in two identical scatterers. However, we would like to clarify that the $N=2$ case, i.e., two identical scatterers, is only a special case in our proposed ZIM-based BICs, which can support a symmetric state and an anti-symmetry state. In that case, the anti-symmetry state (BIC mode) featured with exactly opposite radiated fields can "conceptually" function as a dipole mode with respect from the "global" symmetry, as explained by the reviewer "*for $N=2$ scatterers, the global symmetry is the one of a dipole*".

While this understanding would not be applied to other cases, including two different scatters and more scatters ($N \geq 3$). For example, two scatterers can be designed with different shapes, sizes and filling materials. They can both support the lowest monopole modes, and the related radiative fields at the boundaries will not have any symmetry, as seen in Fig. R1b (Fig.S7-8). In fact, the mode of each object can be high-order monopole modes. As seen in Fig. R1d, the square scatter supports the lowest monopole mode, and the cylindrical one supports high-order monopole mode. The monopole fields would not be an anti-symmetry model of a dipole not only at the boundaries but insider the scatters. Consequently, the explanation of global symmetry will not be accurate. Furthermore, for the case of $N \geq 3$ scatters,

please kindly see Fig. 4b, where two BIC modes appear in a ZIM host with three identical objects. It is very difficult to explain the generated BICs from the view of the "conceptually" dipole model or other analogs. However, the underlying physics of these BICs in all cases, including previous works (higher-order modes in a single scatterer), could be understood from our proposed way, i.e., **the zero total magnetic flux in all scatterers**. Therefore, our proposed ZIM-based-BIC model is new and general, with the related physics and finding beyond the scope of previous knowledge. We believe that our findings expand the family of BICs beyond the founded ones like symmetry-protected BIC, Fabry-Perot BIC and Friedrich-Wintgen BIC.

Fig. R1 Eigenmode analysis for a square object and a cylindrical object embedded in ZIM environment. **a** and **b** represent a radiation mode and a BIC mode, respectively. Two objects support the lowest-order monopole mode. The radius of a cylindrical object is 8 mm, and the side length of a square object is 14.780 mm. **c** and **d** represent a radiation mode and a BIC mode, respectively. The square object supports the lowest-order monopole mode, but the cylindrical object supports high-order monopole mode. The radius of a cylindrical object is 8 mm, and the side length of the square object is 6.437 mm. For all cases, the magnetic field is asymmetric for two objects.

As to the scattering properties, I still do not see the exact difference of this system and other reported previously. Do not they share a Fano-type lineshape near the quasi-BIC state?

Reply: We thank the Reviewer for raising the confusing point. As our proposed system is a new and special one, i.e., ZIM background doped with multiple scatters, the induced BIC comes from the destructive interference of monopole modes, which makes it fundamentally different from previously reported BICs, such as symmetry-protected BIC, accidental BIC, Fabry-Perot BIC and Friedrich-Wintgen BIC. Especially, the typical feature of this system, distinguished from others, is the quasi-BIC with Electromagnetically induced transparency (EIT) like lineshape, which is a special case of Fano resonance [see review article: Nat. Photon. 11, 543–554 (2017)].

To clearly explain this confusing point, we added a sentence and cited the paper to clarify it, reading as,

"The typical feature of our system different from others is that our revealed quasi-BIC shares EIT-like lineshape, which is a special type of Fano resonance⁴¹."

Finally, independent of any technical aspects, I cannot agree with sentences in the introduction of the article such as:

"Therefore, how to achieve robust BICs in a system with no exact geometrical symmetry is still an open question." This sentence is completely misleading as the solution proposed here to solve the problem is not new and is very well understood.

Reply: We thank the Reviewer for such a comment. This sentence aims to state the deficiency of BICs in the symmetry-protected system. The exact geometrical symmetry is very important to achieve a BIC mode and indicate it is meaningful to implement BIC without geometrical symmetry. Considering previous works of ZIM-based BIC with higher-order cavity mode, this description might be inappropriate. However, we have indicated the significant difference between the previous ZIM-based system and our proposed model in the following part of the introduction. Moreover, our proposed BIC is implemented by the ZIM host with randomly placed scatters, which not only has no geometrical symmetry but is a disordered system. By considering these factors, we slightly change this sentence, reading as,

"Therefore, it is significant to achieve robust BICs beyond the limit of geometrical symmetry and how to achieve robust BICs in a disordered system is still an open question."

Reviewers' Comments:

Reviewer #3:

Remarks to the Author:

I read the reply of the authors and the revised article.

The reply of the authors confirms that the modes observed are not globally monopoles. The authors point out that their modes do not have necessarily the dipolar symmetry and I certainly agree. That is not surprising as it is well known that bound states in ENZ-type systems can be formed by any combination of multipoles, but the monopole term is forbidden. This restriction obviously also applies to the system discussed in this article.

Response to the reviewer's comments

Reviewer #3 (Remarks to the Author):

I read the reply of the authors and the revised article.

The reply of the authors confirms that the modes observed are not globally monopoles. The authors point out that their modes do not have necessarily the dipolar symmetry and I certainly agree. That is not surprising as it is well known that bound states in ENZ-type systems can be formed by any combination of multipoles, but the monopole term is forbidden. This restriction obviously also applies to the system discussed in this article.

Reply: We are happy to see that the Reviewer is satisfied with our reply for the problem of “global symmetry”. Indeed, benefiting from the quasi-static property of ZIM, BICs could be realized in a flexible way. Different from previous works, our work presents a general theory to achieve BIC in the doped ZIM system, and breaks this limitation of BIC in the system with the monopole term forbidden.